

# The efficacy of IL-6 inhibitor Tocilizumab in reducing severe COVID-19 mortality: a systematic review

Avi Gurion Kaye[1] and Robert Siegel[1,2]

[1] Human Biology, Stanford University, Stanford, CA, USA
[2] Microbiology & Immunology, Stanford University, Stanford, CA, USA

## ABSTRACT

**Background:** In the absence of highly effective antiviral therapies against SARS-CoV-2, it is crucial to counter the known pathophysiological causes of severe COVID-19. Evaluating the efficacy existing drugs may expedite the development of such therapeutics. Severe COVID-19 is largely the result of a dysregulated immune response characterized by lymphocytopenia, neutrophilia and critical hypercytokinemia, or "cytokine storm," which is largely mediated by the cytokine interleukin-6 (IL-6). The IL-6 inhibitor tocilizumab (TCZ) could potentially suppress the effects of the pro-inflammatory cytokine and thereby lower mortality from the disease. This systematic analysis aimed to investigate and synthesize existing evidence for the efficacy of TCZ in reducing COVID-19 mortality.

**Methodology:** PubMed and SearchWorks searches were performed to locate clinical studies with primary data on TCZ treatment for severe COVID-19. Sixteen case-control studies comparing mortality between TCZ and standard of care (SOC) were identified for quantitative synthesis. The systematic analysis was pre-approved through PROSPERO (CRD42020193479).

**Results:** Combined mortality for the TCZ-treated and SOC groups were 26.0% and 43.4% respectively. In all but one of the studies, the odds ratio of mortality from COVID-19 pointed towards lower fatality with TCZ vs the SOC. A combined random effects odds ratio calculation yielded an odds ratio of 0.453 (95% CI [0.376–0.547], $p < 0.001$). Additionally, 18 uncontrolled trials were identified for qualitative analysis producing a raw combined mortality rate of 16.0%.

**Conclusions:** Important caveats to this research include the lack of prospective randomized control trials and the absence of data from the large COVATA study from the published literature. However, results from this systematic analysis of published research provide positive evidence for the potential efficacy of TCZ to treat severe COVID-19, validating the ethical basis and merit of ongoing randomized controlled clinical trials.

Corresponding author
Avi Gurion Kaye,
agkaye@stanford.edu

## INTRODUCTION

Coronavirus Disease 2019 (COVID-19)—caused by the novel Severe Acute Respiratory Syndrome Coronavirus 2 (SARS-CoV-2)—manifests in a broad range of disease severity.

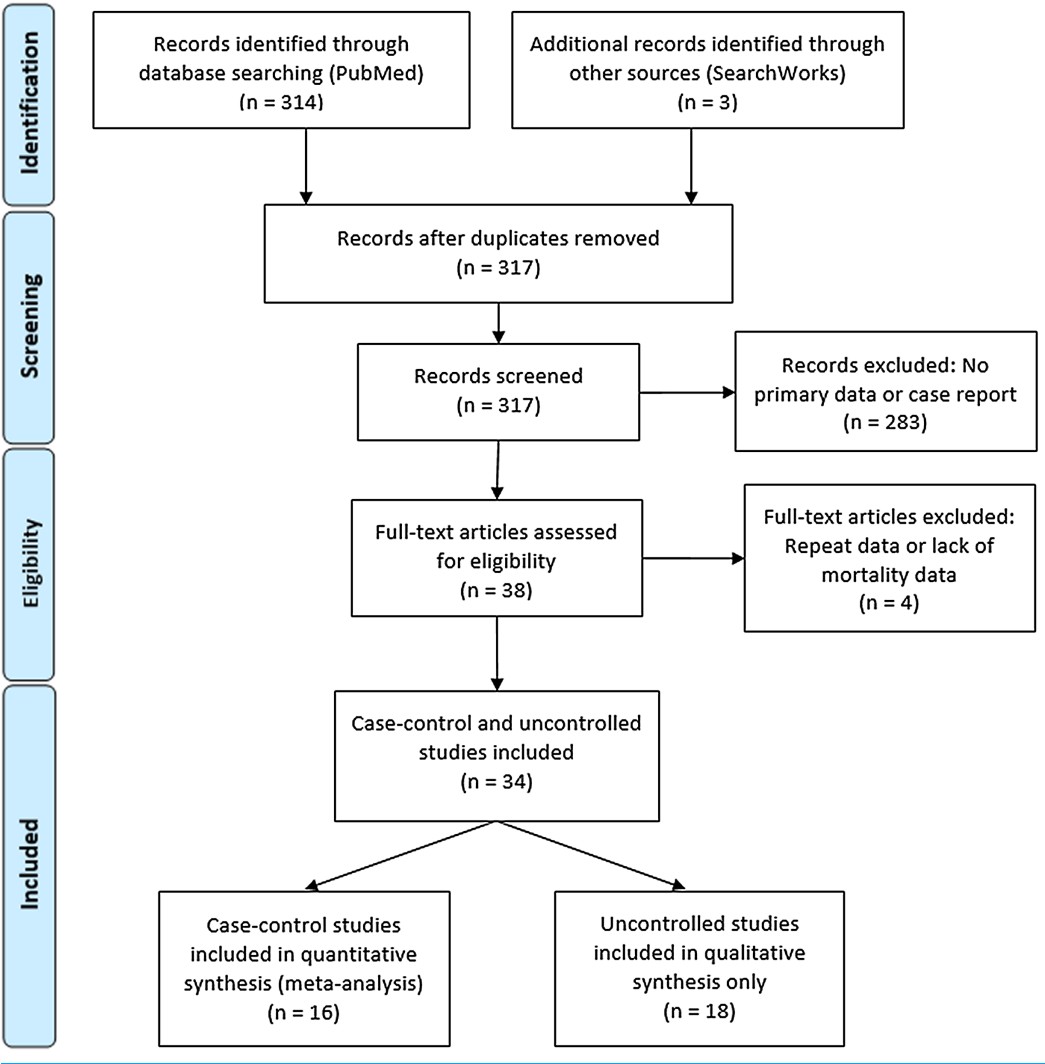

**Figure 1 PRISMA flow diagram for systematic review article search.**

Roughly 85% of confirmed cases present as a mild respiratory illness defined as minor fatigue, low-grade fever and dry cough, 15% develop severe pneumonia requiring hospitalization and 5% become critical indicated by acute respiratory distress syndrome (ARDS), septic shock, and multi-organ failure resulting in ICU admission, mechanical ventilation, and death (*Huang et al., 2020*). The most significant known risk factors for COVID-19-related death are increasing age, chronic comorbidities including diabetes, cardiac disease, pulmonary and kidney dysfunction and male sex (*Docherty et al., 2020*). A dysregulated immune response—characterized by decreased T-cell counts, increased inflammatory cytokines and extra-pulmonary systemic hyperinflammation syndrome—is principally responsible for inducing critical pulmonary failure observed in COVID-19 and largely driven by interleukin-6 (IL-6) (*Siddiqi & Mehra, 2020*). This systematic review (Fig. 1) concerns the efficacy of an interleukin-6 inhibitor, tocilizumab (TCZ) in reducing severe COVID-19 mortality.

## COVID-19 dysregulated immune response and the role of IL-6

Severe COVID-19 is characterized by a dysregulated immune response to SARS-CoV-2 infection that is implicated in disease mortality even after viral load decreases (*Blanco-Melo et al., 2020*). The immune dysregulation presents with two sequential and diametrically opposed reactions that both instigate symptom aggravation (*Cao, 2020*). The first pattern is impaired adaptive immune response with lymphocytopenia, which includes markedly reduced CD4+ and CD8+ T-cells, B-cells and natural killer (NK) cells. While T-cells are significantly decreased in all COVID-19 patients, reduction in B and NK cells are more affected in severe cases (*Shi, 2020*; *Zheng et al., 2020*). Adaptive immune cell depletion impairs the body's ability to clear the virus and mitigate inflammatory reactions (*Jamilloux et al., 2020*).

The second deleterious response is an over-activation of the innate immune system. This pathogenic response is characterized by an increase in neutrophils and pro-inflammatory cytokines including IL-6, IL-1β, IL-2, IL-8, CCL3 and TNF-α. The rapidly increasing cytokine levels—also known as a "cytokine storm"—drives progression to septic shock, tissue damage and multiple organ failure (heart, liver, kidney, respiratory) (*Cao, 2020*). The effects are instigated by excessive NF-κB and JAK/STAT pathway activation, alarmin release by damaged epithelial cells, neutrophil and macrophage infiltration, and alveolar damage by vessel permeability and alveolar wall thickening (*Jamilloux et al., 2020*).

Roughly three-quarters of patients in severe condition present with IL-6 mediated respiratory failure while the other quarter occurs predominately through IL-1β macrophage activation (*Giamarellos-Bourboulis et al., 2020*). IL-6 concentration is thus a reliable predictor of COVID-19 severity as it is significantly elevated in fatal cases (*Zhou et al., 2020*). IL-6 has pleotropic effects including hematopoiesis, metabolic regulation, inflammation, autoimmunity and acute phase response (*Zhang et al., 2020*). Some IL-6-dependent outcomes help stave off infections such as directing neutrophil migration to the infection site, increasing CD8+ T cell cytolytic capacity, and regulating antiviral thermostatic reactions. However, IL-6 is also implicated in viral infection disease progression as it leads to tissue permeability and edema, reduces IFN-γ production, drives anti-apoptotic molecules and promotes excessive neutrophil survival. The above adverse effects promote lethal inflammation and enable viral infiltration to distant organs (*Gubernatorova et al., 2020*). Furthermore, elevated serum IL-6 is associated with impaired cytotoxic activity of NK cells, thus weakening their virus-killing capacity (*Mazzoni et al., 2020*). IL-6 is known to increase the rate of fibrotic clot formation, so it also may play a role in the thrombotic complications observed in COVID-19 (*Gubernatorova et al., 2020*). The renin-angiotensin system, which controls blood pressure and electrolyte balance, is an additional important factor in IL-6 modulation and COVID-19 pathology. As virus binds ACE2, thus reducing its availability, there is an increase of angiotensin II in COVID-19 patients, creating a positive feedback loop that advances pro-inflammatory signaling (*Gubernatorova et al., 2020*). The responses and physiological effects of IL-6 release are summarized in Fig. 2.

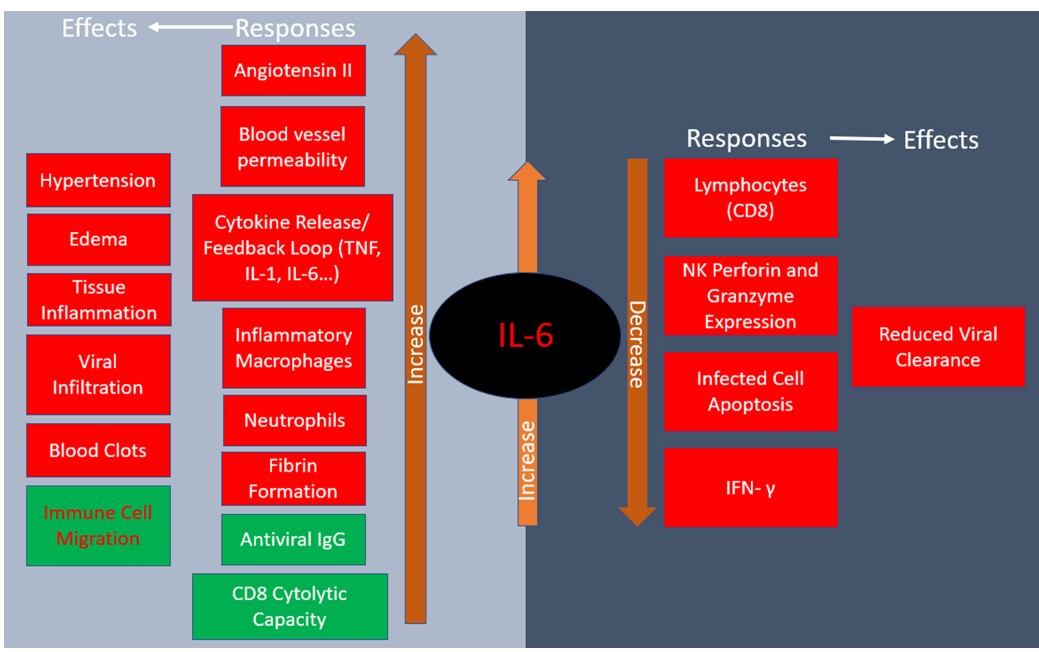

**Figure 2** **Responses to IL-6 release and their physiological effects.** Red indicates negative consequences of IL-6 increase while green designates positive ones in COVID-19.

## COVID-19 treatments

### Antivirals

Without a targeted drug for COVID-19, scientists and clinicians are attempting to rapidly find alternative treatments and solutions to combat the disease's lethal immunological effects (*Mehta et al., 2020*). One strategy is to utilize existing antiviral drugs with the expectation that they may exhibit similar effects against SARS-CoV-2. One of the more promising treatments is RNA-dependent RNA polymerase inhibitor remdesivir which is shown to reduce COVID-19 mortality but is less effective in severe cases (*Beigel et al., 2020*). Other commonly used antivirals are hydroxychloroquine and chloroquine. Despite widespread utilization for COVID-19 patients, evidence is lacking for the clinical efficacy and prophylactic properties of hydroxychloroquine or chloroquine despite their in vitro antiviral and in vivo immunomodulatory properties (*Hernandez et al., 2020*).

### Immune suppressants

There are numerous investigations of existing immune suppressing and anti-cytokine interventions to counter the dysregulated, excessive immune response. After early evidence and recommendations against the use of corticosteroids to treat severe COVID-19 (*Russell, Millar & Baillie, 2020*; *Alhazzani et al., 2020*), a large randomized evaluation of dexamethasone found that the drug significantly reduced 28-day mortality in patients included in the study (rate ratio 0.83; 95% CI [0.74–0.92]; $p < 0.001$). However, mortality rate reductions varied depending on baseline respiratory demands upon randomization as there was reduction for patients on mechanical ventilation and oxygen but not for patients without respiratory support (*Horby et al., 2020*). As of mid-August 2020,

the WHO modified their recommendation against corticosteroids to include judicious administration under respiratory failure with ARDS (*Clinical Management of COVID-19, 2020*). Because of the systemic effects of corticosteroids, more options for targeted immune regulation is warranted. Common targets for inhibition include IL-6, the IL-1 family (IL-1β and IL-18), TNF-α and IFN-γ cytokines and the JAK/STAT pathway (*Jamilloux et al., 2020*).

IL-6 is a particularly promising target due to its correlation with ARDS severity and mortality (*Coomes & Haghbayan, 2020*). IL-6 inhibitors are already successfully utilized for other cytokine storm syndromes such as adverse T cell therapy reactions and Still's disease-associated hemophagocytic lymphohistiocytosis (*Jamilloux et al., 2020*). In COVID-19, IL-6 inhibitors should be carefully administered with appropriate timing due to its control of viral replication (*Velazquez-Salinas et al., 2019*).

### IL-6 inhibitor Tocilizumab

Tocilizumab (TCZ) [Actemra] is a recombinant monoclonal antibody with a humanized murine variable domain and a human IgG1 constant domain. TCZ binds to both membrane-bound and soluble IL-6 receptors, thus preventing IL-6 mediated signal transduction (Fig. 3). The drug was initially developed to treat rheumatoid arthritis and now it is also approved for giant cell arteritis and similar autoimmune ailments. Its safety profile was analyzed in a phase III double-blind placebo-controlled trial and it is reportedly effective in treating other cases severe cytokine release syndrome such as chimeric antigen receptor T-cell immunotherapy (*Zhang et al., 2020*).

While TCZ is not yet approved for treatment of COVID-19, clinicians across the globe are utilizing the drug under emergency use authorization, including in the United States. One of the largest initial observational studies of TCZ evaluated 547 COVID-19 ICU patients in New Jersey comparing the survival rate of 134 individuals treated with standard of care (SOC) and TCZ with SOC controls finding a 46% and 56% mortality rate respectively and a 0.76 adjusted hazard ratio (*Ip et al., 2020*). However, there was insufficient statistical power to conclude clinical efficacy of TCZ with the clinical data and they only focused cases that already progressed to a critical stage leading to an inflated mortality rate for both groups.

Randomized controlled trials (RCT) are the gold standard for evaluating the clinical efficacy of a drug. The first analyzed RCT for TCZ presented negative results. Genentech (Roche)—the producer of TCZ—discontinued their 450-participant phase III trial COVACTA because it failed to meet the primary endpoint of improved clinical status after 4 weeks with TCZ vs the SOC (*Roche, 2020*). The disappointing outcome from COVACTA places serious doubt on the efficacy of TCZ against COVID-19. Additionally, another IL-6 inhibitor carlumab failed its Phase III RCT in the United States but still has some ongoing (NCT04322773, NCT04327388, NCT04412772) (*Clinical Trials Arena, 2020*). Other IL-6 inhibitors under investigation include siltuximab (SYLVANT, NCT04322188) and fingolimod (NCT04280588) Nonetheless, several phase III and phase II RCTs remain in progress into at least September (REMDACTA, NCT04409262; NCT04372186, NCT04356937) and healthcare providers are still currently administering TCZ globally for

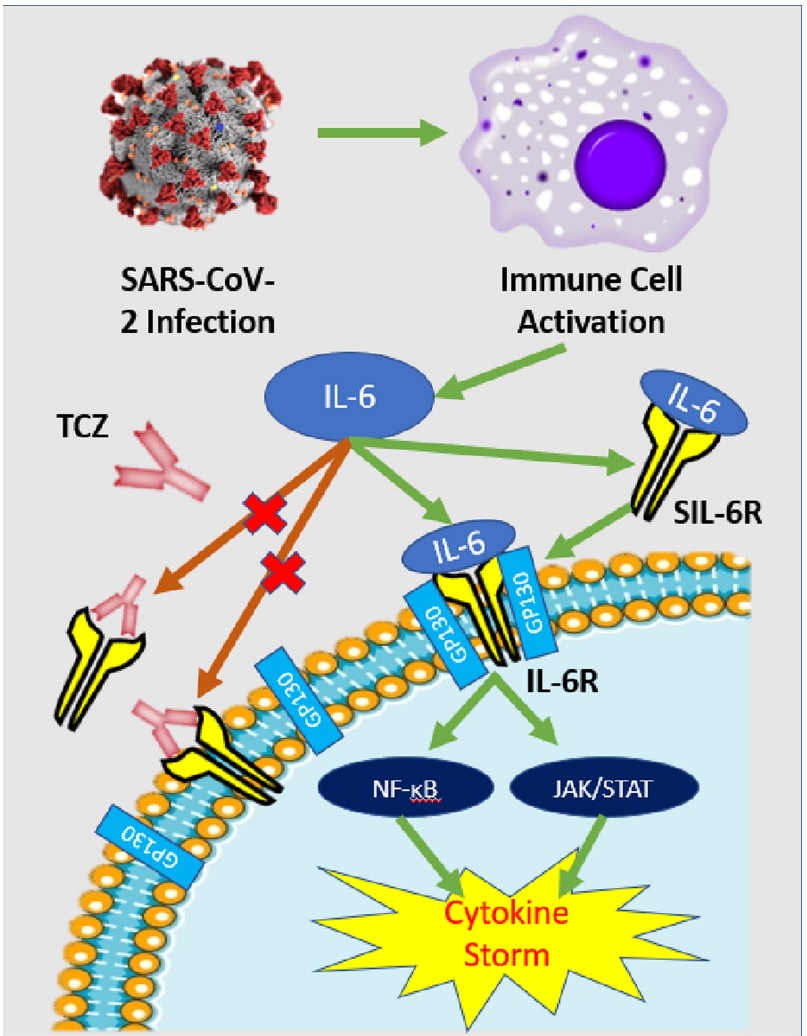

**Figure 3** **IL-6 pathway and TCZ mechanism of action.** Simplified schematic of IL-6 pathway and TCZ mechanism of action. After SARS-CoV-2 infection, immune cells—predominately macrophages—release cytokines including IL-6. IL-6 can either attach to its respective cell receptor (IL-6R) or soluble receptor (SIL-6R) which activates both the NF-κB and JAK/STAT pathways that can induce a cytokine storm (*Coomes & Haghbayan, 2020*). TCZ antibodies can bind to the IL-6R and SIL-6R to prevent signal transduction and reduce the chance of cytokine storm (*Zhang et al., 2020*).

advanced COVID-19 cases. However, the logic of suppressing IL-6 remains convincing. Therefore, it is essential that the known impact of TCZ is analyzed in a systematic manner to ascertain whether its continuing use is ethical, even in RCTs. Since dexamethasone is a broadly acting immunosuppressant, it should be noted that the use of this drug in clinical trials may obscure the effects of more targeted immunomodulatory drugs like TCZ.

The most recent systematic reviews on the use of TCZ for COVID-19 identified clinical trials without data synthesis (*Cortegiani et al., 2020*) and performed a large meta-analysis of controlled trials but did not address issues with the individual studies (*Aziz et al., 2020*). Therefore, this systematic review will synthesize the evidence from individual case-control studies, analyze uncontrolled trials and evaluate their methods to

determine whether the drug is potentially effective at reducing severe COVID-19-related mortality, thus corroborating the logic for continuing RCTs to evaluate the potential use of IL-6 inhibitors.

## SYSTEMATIC REVIEW METHODS AND STATISTICS

Articles utilized for the systematic review were selected from a PubMed search on August 4, 2020. Both authors screened and reviewed each paper, and disagreements were resolved by a third reader. Data was independently extracted by the readers. For the initial screening, the primary search terms were "COVID-19" or "SARS-CoV-2" and "tocilizumab." Papers with primary data for a case-control study comparing mortality rate from severe COVID-19 between TCZ and standard of care (SOC) were included for data synthesis. Uncontrolled studies on severe COVID-19 mortality with TCZ were reviewed separately without data synthesis. Exclusion criteria included papers without primary data, case reports, reviews, protocols, and studies without mortality numbers available or patient data that may be used in another studies. An additional search was performed on SearchWorks to identify case-control studies not found in PubMed.

For each study included in the synthesis, the mortality rate for the TCZ and SOC group were calculated. In the controlled studies, the odds ratio (RR) of mortality from COVID-19 with TCZ vs the SOC was determined followed by the 95% confidence interval (CI) and $p$-value calculation. The data from the individual controlled studies were synthesized by a random effects meta-analysis calculation using MedCalc software. MedCalc was also used to perform a sample size calculation with an alpha of 0.01 and power of 90% to detect a difference between the total crude TCZ and SOC mortality rates.

The systematic review protocol was pre-registered with PRISMA and approved on June 22, 2020 (CRD42020193479).

## RESULTS

A total of 314 articles were identified by the initial PubMed search and three additional case-control studies were found on a SearchWorks (Fig. 1) (*Moher et al., 2009*). A total of 38 articles were selected for full-text review yielding 16 uncontrolled studies for qualitative analysis and 18 case-control studies for both quantitative synthesis and qualitative analysis. The study characteristics for the controlled studies are summarized in Table S1 while the uncontrolled studies are outlined in Table S3.

### Controlled studies

The systematic review of sixteen controlled studies encompassed a total of 1,008 TCZ-treated and 1,537 SOC control patients (Table 1). A total of 13 of the studies occurred in a single medical center while the remaining three aggregated data from multiple hospitals. The largest patient contributions to the analysis were from the multiple-hospital studies (*Ip et al., 2020*; *Guaraldi et al., 2020*) The baseline patient characteristic for all but one study (*Guaraldi et al., 2020*) was severe COVID-19, generally qualified by oxygen supplementation needs. *Ip et al. (2020)*, *Eimer et al. (2020)* and *Potere et al. (2020)* only analyzed patients who were already admitted into the ICU. A total of 61% and 44.8% of

**Table 1** Quantitative synthesis of individual case-control study mortality data.

| Study | TCZ mortality | Controls | Odds ratio | 95% CI | Weight % (random effects) | *p* Value |
|---|---|---|---|---|---|---|
| Klopfenstein et al. (2020) | 4/20 (20%) | 12/25 (48%) | 0.271 | [0.0704–1.042] | 3.54 | 0.0575 |
| Campochiaro et al. (2020) | 5/32 (15.6%) | 11/33 (33.3%) | 0.37 | [0.112–1.227] | 4.23 | 0.104 |
| Capra et al. (2020) | 2/62 (3.2%) | 11/23 (47.8%) | 0.0364 | [0.00713–0.185] | 2.61 | 0.0001 |
| Colaneri et al. (2020) | 5/21 (23.8%) | 6/21 (28.6%) | 0.781 | [0.197–3.106] | 3.41 | 0.726 |
| Rojas-Marte et al. (2020) | 50/96 (52.1%) | 60/97 (61.9%) | 0.67 | [0.378–1.189] | 9.7 | 0.171 |
| Wadud et al. (2020) | 17/44 (38.6%) | 26/50 (52%) | 0.581 | [0.255–1.323] | 6.9 | 0.196 |
| Ip et al. (2020) | 62/134 (46.3%) | 231/413 (55.9%) | 0.678 | [0.459–1.003] | 12.2 | 0.0519 |
| Roumier et al. (2020) | 3/30 (10%) | 9/30 (30%) | 0.259 | [0.0623–1.079] | 3.24 | 0.0635 |
| Guaraldi et al. (2020) | 13/179 (7.3%) | 73/365 (20%) | 0.309 | [0.2166–0.574] | 9.11 | 0.0002 |
| Patel et al. (2020) | 11/42 (26.2%) | 11/41 (28.6%) | 0.968 | [0.355–2.565] | 5.62 | 0.947 |
| Eimer et al. (2020) | 4/22 (18.2%) | 7/22 (31.8%) | 0.476 | [0.116–1.94] | 3.31 | 0.301 |
| Canziani et al. (2020) | 17/64 (26.6%) | 24/64 (37.5%) | 0.603 | [0.285–1.277] | 7.61 | 0.187 |
| Gokhale et al. (2020) | 33/70 (47.1%) | 61/91 (67%) | 0.345 | [0.185–0.644] | 8.85 | 0.0008 |
| Rossotti et al. (2020) | 20/74 (26.4%) | 86/146 (58.9%) | 0.517 | [0.301–0.887] | 9.23 | 0.0166 |
| Somers et al. (2020) | 14/78 (17.9%) | 22/76 (35.5%) | 0.379 | [0.189–0.836] | 7.67 | 0.0151 |
| Potere et al. (2020) | 2/40 (5%) | 12/40 (30%) | 0.123 | [0.0254–0.593] | 2.76 | 0.009 |
| Total (random effects) | 262/1008 (26.0%) | 667/1537 (43.4%) | 0.453 | [0.376–0.547] | 100 | <0.001 |

both cases and controls in *Rojas-Marte et al. (2020)* and *Roumier et al. (2020)* respectively began the trial in critical condition. There was some variation in TCZ administration and SOC treatments, the most common being hydroxychloroquine—utilized in all but one study (*Capra et al., 2020*)—and lopinavir/ritonavir. Length of observation ranged from 7 days to 30 days with endpoints of death or discharge. Mean age of participants in the treatment and control groups was 55.5–76.8 with no more than 6.1 years separating the two groups within one study.

Combined mortality for the TCZ-treated and SOC groups were 26.0% and 43.4% respectively. All of the studies trended toward lower mortality from severe COVID-19 with TCZ vs the SOC with the exception of *Patel et al. (2020)* which showed no benefit. Six studies yielded a statistically significant result (*Capra et al., 2020*; *Guaraldi et al., 2020*; *Gokhale et al., 2020*; *Rossotti et al., 2020*; *Somers et al., 2020*; *Potere et al., 2020*) (Fig. 4). A random effects odds ratio analysis generated an odds ratio of 0.453 (95% CI [0.376–0.547]) with a *p*-value less than 0.001. A sample size analysis with alpha of 0.01 and power of 90% determined that 392 total case and control patients are needed to detect a difference between 26.0% and 43.4% mortality. With the exception of *Capra et al. (2020)*, the studies also scattered symmetrically around the overall odds ratio from the analysis suggesting a low likelihood of publication bias (Fig. 5). TCZ patients in five studies had significant secondary bacteremia but one reported a lower rate than the SOC (*Rojas-Marte et al., 2020*). Eight studies reported no adverse effects from TCZ.

No pattern was identified in terms of the length of hospitalization between TCZ and SOC. Nine studies found a lower rate of ICU admission with TCZ, four with statistical

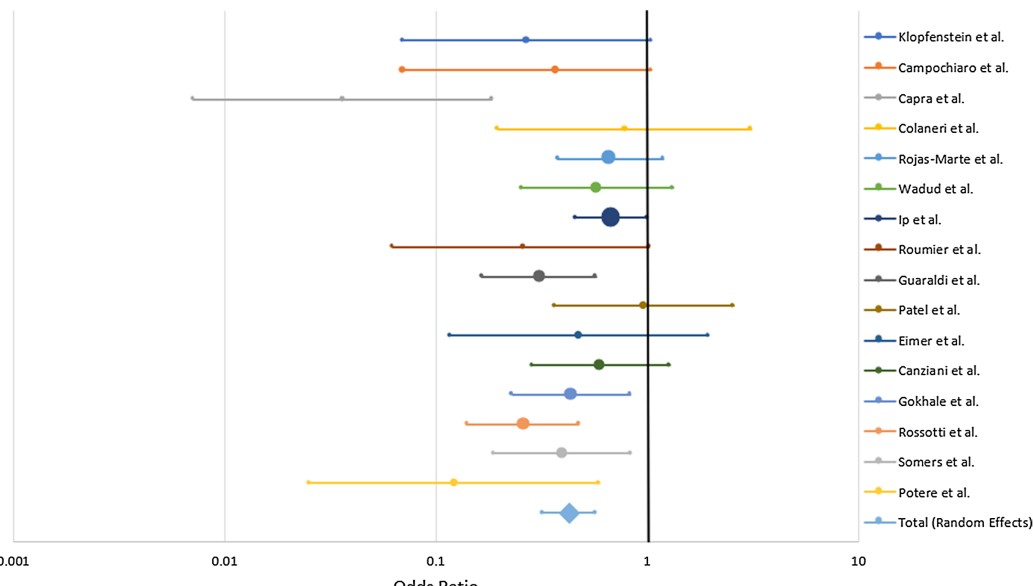

**Figure 4 Odds ratio of mortality from severe COVID-19 between TCZ and standard of care.** Forrest plot depicting odds ratio for death from severe COVID-19 with TCZ vs the SOC. Center point on each line is the odds ratio for the study with the size of the circle correlating to the percent contribution to the total random effects calculation. Horizontal length corresponds to the 95% CI. The total (random effects) synthesizes data from the 16 individual case-control studies.

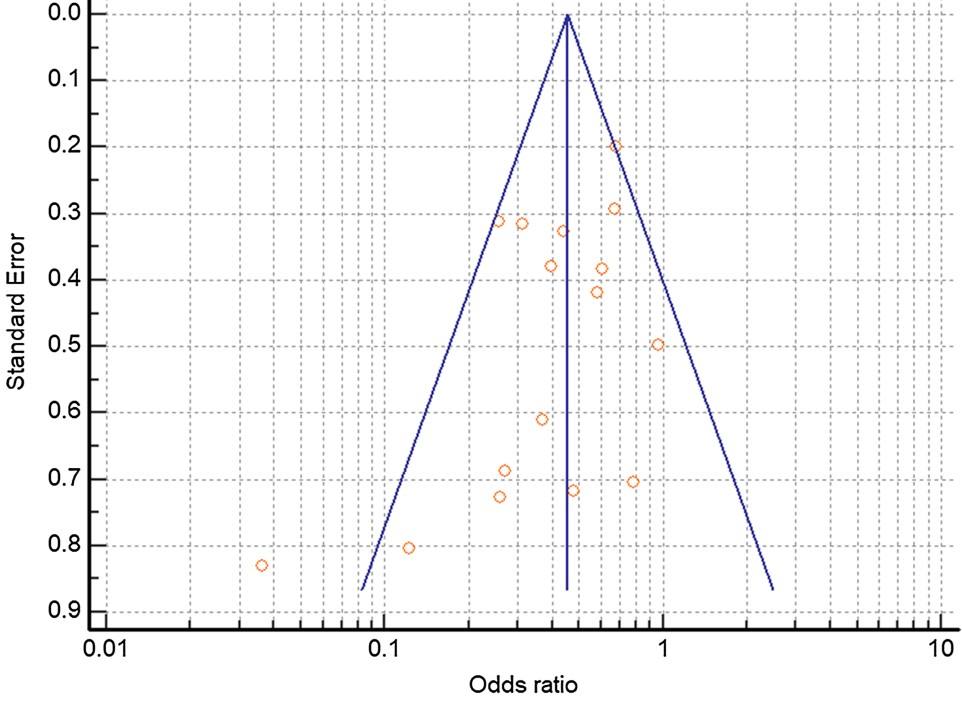

**Figure 5 Funnel plot of controlled trials to detect potential publication bias.** Funnel plot relating the odds ratio to the standard of the effect estimate for each study. Vertical blue line depicts the overall random effects odds ratio.

**Table 2 Comparison of mortality rate between controlled (*n* = 14) and uncontrolled (*n* = 18) studies.** Mortalities and total patient values are simple tallies from each study.

|  | Mortalities | Total | Mortality rate (%) | SD (%) |
|---|---|---|---|---|
| Controlled | 150 | 778 | 19.3 | 11.9 |
| Uncontrolled | 141 | 886 | 16.0 | 9.87 |
| Difference | 3.3% |  |  |  |
| 95% CI | [−10.91 to 4.31] |  |  |  |
| Significance Value | *p* = 0.384 |  |  |  |

significance (*Colaneri et al., 2020*; *Roumier et al., 2020*; *Guaraldi et al., 2020*; *Canziani et al., 2020*) (Table S1). Only one study (*Colaneri et al., 2020*) provided both baseline and post-treatment values for biomarkers such as IL-6, CRP, lymphocytes, neutrophils, ALT etc.

Several variables varied between studies that could impact the mortality rate results. These include SOC, observation time, TCZ administration, treatment date, baseline clinical characteristics, geographic location and resources and mean/median age. Study design was also an important varying factor that may change results as 9 of the 16 studies matched cases with controls and 13 studies were retrospective as opposed to prospective cohort.

### Uncontrolled studies

The 18 uncontrolled trial (Table S3) encompassed 886 total patients who received TCZ. The mortality rate from severe COVID-19 ranged from 0% to 42.4% (SD 9.87%), although the two studies with 0% had relatively small sample sizes (20 for *Xu, Han & Li (2020)* and 12 for *Borku Uysal et al. (2020)*). The raw overall mortality rate from the 12 studies is 16.0%. The initial patient severity level ranged from "severe"—requiring supplemental oxygen—to ICU admission. One study only investigated ICU patients (*Issa et al., 2020*) and others included as many as 77.7% on MV. SOC varied more widely in the uncontrolled trials than the controlled, but hydroxychloroquine was still the most common additional drug used. Few major side effects such as bacterial/fungal infections and increased hepatic enzymes were reported.

Comparing the combined data between uncontrolled (*n* = 18) and controlled trials (*n* = 14) with TCZ (Table 2), excluding controlled studies with all the patients initially in the ICU, the mortality rate was 19.3% and 16.0% respectively (*p* = 0.384).

## DISCUSSION

### Controlled studies

The purpose of this systematic analysis was to analyze and synthesize clinical data on the efficacy of TCZ treatment against severe COVID-19. In the sixteen existing case-control trials published as of August 4, 2020, combined mortality for the TCZ-treated and SOC groups were 26.0% and 43.4% respectively. All of the studies at least trended towards reduced mortality with TCZ-with one exception (*Patel et al., 2020*) that showed no

benefit—(Fig. 4) including six with statistical significance (*Capra et al., 2020*; *Guaraldi et al., 2020*; *Gokhale et al., 2020*; *Rossotti et al., 2020*; *Somers et al., 2020*; *Potere et al., 2020*). After performing a quantitative synthesis, the random effects odds ratio of mortality with TCZ vs the SOC was 0.453 (95% CI [0.376–0.547], $p < 0.001$) illustrating a reported difference in patient outcomes associated with the use of TCZ. There was no indication of publication bias except for *Capra et al. (2020)* (Fig. 5) and the number of patients in the meta-analysis exceeded the 392 required total case and control patients to statistically detect the difference in mortality rate. Some studies also secondarily investigated the rate of ICU admission with nine finding a lower rate with TCZ vs SOC and four having statistical significance (*Colaneri et al., 2020*; *Roumier et al., 2020*; *Guaraldi et al., 2020*; *Canziani et al., 2020*) (Table S1). Although the value of any single controlled clinical study does not hold definitive proof of efficacy, the consistent positive trend and statistical significance from the combined data in this analysis corroborates TCZ's potential positive effects.

TCZ also appears to be mostly benign as few serious side effects were reported in most of the studies. As expected from an immune suppressant, the most notable adverse event was secondary infection reaching as high as 32.4% of participants (*Rossotti et al., 2020*) but not at a statistically greater rate than SOC.

It is important to acknowledge that only one of the studies (*Guaraldi et al., 2020*) randomized who received TCZ which introduces the possibility for selection bias into each study's research methodology. Additionally, only one study provided both baseline and post-treatment values for important biomarkers, notably IL-6, CRP, neutrophils and lymphocytes. Specific strengths and shortcomings for the controlled trials are outlined in Table S2, which points towards additional methodological issues in each of the studies analyzed. Common themes include short observation timing, small sample size, difference in treatment time (later patients may benefit from better SOC), variation in disease severity within groups which can confound results, and changes to TCZ administration mid-study. Future clinical trials can be improved by addressing the methodological issues outlined above.

Additionally, it is possible that individual studies finding a negative effect of TCZ were not published and could not be accounted for in this systematic review. Most notably, the known public results from the COVACTA trial were not yet released in published format as of late August and were therefore not included in the quantitative synthesis. It is significant that the RCT COVACTA did not find reduced COVID-19 mortality with TCZ vs SOC, placing doubt as to the drug's potential efficacy and the ethical basis of continuing other trials. Nonetheless, there are still multiple RCTs in progress to evaluate the efficacy of TCZ (NCT04409262; NCT04372186, NCT04356937, NCT04412772). Data from this systematic review adds merit to clinical investigations despite the initial negative results from COVACTA.

## Uncontrolled studies

Uncontrolled trials were analyzed separately to explore trends in treatment data. Recognizing the wide variation in patient outcomes between the studies, the combined

mortality rate from 12 single-arm studies using TCZ against severe COVID-19 was 16.0% (SD 9.87%). However, without clinical trials, it is difficult to determine the baseline rate COVID-19 mortality for hospitalized patients to compare with the rate calculated from the uncontrolled trials. Clinical data reviews are subject to geographic and demographic differences, such as a 5,700-person evaluation in New York from April 2020 that found a 21% fatality rate for hospitalized patients which is higher than other locations and later studies (*Richardson et al., 2020*). Participants must be matched to controls to eliminate bias and account for other confounding factors to draw definitive conclusions. Comparing the uncontrolled and controlled trials from this systematic review, excluding controlled trials with all the patients initially in the ICU (*Ip et al., 2020*; *Rojas-Marte et al., 2020*), there was no significant difference in mortality rate between the two experimental approaches ($p = 0.384$). This observation supports the assertion that the combined reported results from the uncontrolled results from this systematic analysis are potentially accurate and corroborate the merit in evaluating TCZ efficacy. Nevertheless, the uncontrolled trials should still be evaluated with some degree of skepticism. Specific shortcomings for the individual uncontrolled trials are delineated in Table S4.

### TCZ treatment timing

There is a question of timing for the IL-6 blocking treatment. All of the studies evaluated included only patients who were already in a severe disease state. Given the patterns of COVID-19 pathology and immune dysregulation, it may be logical to defer TCZ until the inflammatory phase due to the positive effects of IL-6 release in the acute infection stage which theoretically prevents SARS-CoV-2 proliferation. Given the unique, aberrant immune reaction in COVID-19, some researchers argue that the optimal time to employ targeted immune suppressants such as TCZ, in order to curtail and not enhance mortality, is when patients begin to trend towards hypoxia and inflammation (*Siddiqi & Mehra, 2020*). However, this timing is theoretical and must be demonstrated in a RCT.

### Limitations

There are notable limitations to this systematic analysis and qualitative synthesis of TCZ. First, only one of the studies presented randomized who received TCZ, opening the possibility for selection bias and confounding factors that cannot be accounted for statistically. This systematic analysis synthesizes data from studies with different SOCs, geographies, resources, demographics, and TCZ dosing amount, number and timing. It is not possible to conclude that TCZ is efficacious in reducing COVID-19 mortality, simply that the data trends towards a lower odds ratio for mortality with incomplete generalizability. Similarly, while patients across all of the studies were at least in severe condition, the combined data still represents individuals at various stages of COVID-19. Not all of the studies offered a longitudinal time component, so an overall hazard ratio or Kaplan–Meir survival curve cannot be produced. Additionally, many patients were still in the hospital at the end of the observation period potentially skewing the mortality rate. Importantly, in the random effects odds ratio calculation, there was no control for age, sex and baseline characteristics like individual studies were able to accomplish. As noted,

the uncontrolled trials on TCZ cannot be adequately evaluated without direct comparison to a control group. Finally, it is possible that studies finding a negative effect of TCZ were not published and not accounted for in this systematic review.

## CONCLUSIONS

A systematic review of the clinical data of IL-6 inhibitor tocilizumab (TCZ) for severe COVID-19 points towards efficacy in reducing mortality from the disease. There are numerous notable methodological limitations in the studies analyzed including the lack of randomization in controlled trials and potential for an inadequate evaluation due to unpublished data. However, the results from this systematic analysis corroborate the logic and ethical basis for ongoing phase III RCTs on TCZ. In light of this analysis, several factors would facilitate the evaluation of TCZ as a therapeutic for the immune dysregulation associated with COVID-19: (1) Publication of the results from unpublished clinical trials, (2) Completion of additional randomized controlled trials especially where the potentially complicating effects of dexamethasone may be ruled out, (3) Comparison of TDZ findings with results assessing other IL-6 inhibitors such as sarilumab or siltuximab, and (4) Completion of additional metabolic studies measuring the levels of immune mediators and biomarkers in SARS-CoV-2 infected animal models and humans treated with TCZ. It will also be useful to assess the possibility of therapeutic synergy between antiviral agents such as remdesivir and TCZ. The use of TCZ outside the clinical trial context is discouraged until results from these clinical trials are released.

### Funding
The authors received no funding for this work.

### Competing Interests
The authors declare that they have no competing interests.

### Author Contributions
- Avi Gurion Kaye conceived and designed the experiments, performed the experiments, analyzed the data, prepared figures and/or tables, authored or reviewed drafts of the paper, and approved the final draft.
- Robert Siegel conceived and designed the experiments, performed the experiments, analyzed the data, authored or reviewed drafts of the paper, and approved the final draft.

### Data Availability
  The raw data from this systematic review are available in Table 1 and Tables S1 and S3.

### Supplemental Information
Supplemental information for this article can be found online at http://dx.doi.org/10.7717/peerj.10322#supplemental-information.

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

## FURTHER READING

**Alattar R, Ibrahim T, Shaar S, Abdalla S, Shukri K, Daghfal J, Khatib M, Aboukamar M, Abukhattab M, Alsoub H, Almaslamani M, Omrani A. 2020.** Tocilizumab for the treatment of severe coronavirus disease 2019. *Journal of Medical Virology* **92(10)**:2042–2049 DOI 10.1002/jmv.25964.

**Campins L, Boixeda R, Perez-Cordon L, Aranega R, Lopera C, Force L. 2020.** Early tocilizumab treatment could improve survival among COVID-19 patients. *Clinical Experimental Rheumatology* **38(3)**:578.

**Fernández-Ruiz M, López-Medrano F, Asín M, De la Calle M, Bueno H, Caro-Teller J, Catalan M, De la Calle C, Garcia R, Gomez C, Laguna-Goya R, Lizasoain M, Martinez-Lopez J, Origuen J, Pablos J, Ripoll M, Juan R, Trujillo H, Lumbreras C, Aguado J. 2020.** Tocilizumab for the treatment of adult patients with severe COVID-19 pneumonia: a single-center cohort study. *Journal of Medical Virology* **38**:529 DOI 10.1002/jmv.26308.

**Jordan S, Zakowski P, Tran H, Smith E, Gaultier C, Marks G, Zabner R, Lowenstein H, Oft J, Bluen B, Le C, Shane R, Ammerman N, Vo A, Chen P, Kumar S, Toyoda M, Ge S, Huang E. 2020.** Compassionate use of tocilizumab for treatment of SARS-CoV-2. *Clinical of Infectious Diseases* **382**:2012 DOI 10.1093/cid/ciaa812.

**Knorr J, Colomy V, Mauriello C, Ha S. 2020.** Tocilizumab in patients with severe COVID-19: a single-center observational analysis. *Journal of Medical Virology* **92(11)**:2813–2820 DOI 10.1002/jmv.26191.

**Luo P, Liu Y, Qiu L, Liu X, Liu D, Li J. 2020.** Tocilizumab treatment in COVID-19: a single center experience. *Journal of Medical Virology* **92(7)**:814–818 DOI 10.1002/jmv.25801.

**Morena V, Milazzo L, Oreni L, Bestetti G, Fossali T, Bassoli C, Torre A, Cossu M, Minari C, Ballone E, Perotti A, Mileto D, Niero F, Merli S, Foschi A, Vimercati S, Rizzardini G, Sollima S, Bradanini L, Galimberti L, Colombo R, Micheli V, Negri C, Ridolfo A, Meroni L, Galli M, Antinori S, Corbellino M. 2020.** Off-label use of tocilizumab for the treatment of SARS-CoV-2 pneumonia in Milan, Italy. *European Journal of Internal Medicine* **76**:36–42 DOI 10.1016/j.ejim.2020.05.011.

**Moreno-Pérez O, Andres M, Leon-Ramirez JM, Sanchez-Paya J, Rodriguez J, Sanchez R, Garcia-Sevila R, Boix V, Gil J, Merino E. 2020.** Experience with tocilizumab in severe COVID-19 pneumonia after 80 days of follow-up: a retrospective cohort study. *Journal of Autoimmunity* **114**:102523 DOI 10.1016/j.jaut.2020.102523.

**Patel A, Shah K, Dharsandiya M, Patel K, Patel T, Patel M, Reljic T, Kumar A. 2020.** Safety and efficacy of tocilizumab in the treatment of severe acute respiratory syndrome coronavirus-2 pneumonia: a retrospective cohort study. *Medical Microbiology* **38(1)**:117 DOI 10.4103/ijmm.IJMM_20_298.

**Price C, Altice F, Shyr Y, Koff A, Pischel L, Goshua G, Azar M, Mcmanus D, Chen S, Gleeson S, Britto C, Azy V, Kaman K, Gaston D, Davis M, Burrello T, Harris Z, Villanueva M, Aoun-Barakat L, Kang I, Seropian S, Chupp G, Bucala R, Kaminski N, Lee A, LoRusso P, Topal J, CelaCruz C, Malinis M. 2020.** Tocilizumab treatment for cytokine release syndrome in hospitalized COVID-19 patients: survival and clinical outcomes. *Chest* **158(4)**:1397–1408 DOI 10.1016/j.chest.2020.06.006.

**Quartuccio L, Sonaglia A, McGonagle D, Fabris M, Peghin M, Pecori D, De Monte A, Bove T, Curcio F, Bassi F, De Vita S, Tascini C. 2020.** Profiling COVID-19 Pneumonia progressing into the cytokine storm syndrome: results from a single Italian Centre study on tocilizumab versus standard of care. *Journal of Clinical Virology* **129**:104444 DOI 10.1016/j.jcv.2020.104444.

**Sciascia S, Apra F, Baffa A, Baldovino S, Boaro D, Boero R, Bonora S, Calcagno A, Cecchi I, Cinnirella G, Converso M, Cozzi M, Crosasso P, De Iaco F, Di Perri G, Eandi M, Fenoglio R, Giusti M, Imperiale D, Imperiale G, Livigni S, Manno E, Massara C, Milone V, Natale G, Navarra M, Oddone V, Osella S, Piccioni P, Radin M, Roccatello D, Rossi D. 2020.** Pilot prospective open, single-arm multicentre study on off-label use of tocilizumab in patients with severe COVID-19. *Clinical and Experimental Rheumatology* **38**:529–532.

**Strohbehn G, Heiss B, Rouhani S, Trujillo J, Yu J, Kacew A, Higgs E, Bloodworth J, Cabanov A, Wright R, Koziol A, Weiss A, Danahey K, Karrison T, Edens C, Ventura I, Pettit N, Patel B, Pisano J, Strek M, Gajewski T, Ratain M, Reid P. 2020.** COVIDOSE: low-dose tocilizumab in the treatment of Covid-19. *medRxiv* DOI 10.1101/2020.07.20.20157503.

**Tomasiewicz K, Piekarska A, Stempkowska-Rejek J, Serafinska S, Gawkowska A, Parczewski M, Niscigorska-Olsen J, Lapinski T, Zarebska-Michaluk D, Kowalska J, Horban A, Flisiak R. 2020.** Tocilizumab for patients with severe COVID-19: a retrospective, multi-center study. *Expert Review of Anti-infective Therapy* **130**:1–8 DOI 10.1080/14787210.2020.1800453.

**Toniati P, Piva S, Cattalini M, Garrafa E, Regola F, Castelli F, Franceschini F, Airo P, Bazzani C, Beindorf E, Berlendis M, Bezzi M, Bossini N, Castellano M, Cattaneo S, Cavazzna I, Contessi G, Crippa M, Delbarba A, De Peri E, Faletti A, Filippini M, Filippini M, Frassi M, Gaggiotti M, Gorla R, Lanspa M, Lorenzotti S, Marino R, Maroldi R, Metra M, Matteelli A, Modina D, Moioli G, Montani M, Muiesan M, Odolini S, Peli E, Pesenti S, Pezzoli M, Pirola I, Pozzi A, Proto A, Rasulo F, Renisi G, Ricci C, Rizzoni D, Romanelli G, Rossi M, Salvetti M, Scolari F, Signorini L, Taglietti M, Tomasoni G, Tomasoni L, Turla F, Valsecchi A, Zani D, Zuccala F, Zunica F, Foca E, Andreoli L, Latronico N. 2020.** Tocilizumab for the treatment of severe COVID-19 pneumonia with hyperinflammatory syndrome and acute respiratory failure: a single center study of 100 patients in Brescia, Italy. *Autoimmunity Reviews* **19(7)**:102568 DOI 10.1016/j.autrev.2020.102568.