# Peer review of "The efficacy of IL-6 inhibitor Tocilizumab in reducing severe COVID-19 mortality: a systematic review"

_PeerJ, doi:10.7717/peerj.10322_

## Round 0.1 · original submission · Minor Revisions

Dear Dr. Kaye,

Your manuscript was evaluated by three independent reviewers. The reviewers find the manuscript to be of potential interest though they have comments on the manuscript. Please revise the manuscript based on their suggestions.

Sincerely,

Gunjan

·

Basic reporting

1. The language used is error-free and unambiguous, conveying the necessary scientific details.
2. Reference list is clear, correct and sufficient. Some comments listed below would need more references to be added, but the current ones are ideal for the written content.
3. The authors have mentioned how 3/4th of the severe cases present with IL6 mediated respiratory failure. Alluding to the remaining 1/4th cases (with appropriate references) would be helpful.
4. A schematic showing the mechanism of action of TCZ and its downstream effects would be useful.
5. Please clarify what “potentially repeating patient data” indicates.

Experimental design

6. This being a review article, it addresses all the relevant clinical trials and primary research articles to arrive to scientific conclusions about efficacy of TCZ in COVID-19 patients.
7. Are there any trials of TCZ on younger patients suffering from COVID-19? If so, since they weren’t in this review analysis, what were the results and maybe mentioning those in the discussion section would be useful.
8. Why did the one mentioned study derive results showing increased mortality with TCZ treatment compared to SOC? Please elaborate.

Validity of the findings

COVID-19 affected individuals and results/conclusions arising from the studies are still in nascent phases and ever-changing based on recent research.
Hence even though this review suggests that TCZ could be a potentially beneficial treatment for patients, the authors have highlighted the caveats of the research.
Data is statistically analysed and speculations have been clearly outlined.

Additional comments

The manuscript is a great attempt at bringing together all the relevant research for IL6 inhibitors, especially TCZ in COVID treatment. It would be of importance to the field, as we slowly know more about this virus and its deleterious effects.

Reviewer 2 ·

Basic reporting

No comments

Experimental design

No comments

Validity of the findings

No comments

Additional comments

Detailed systematic review!

Reviewer 3 ·

Basic reporting

No comment

Experimental design

No comment

Validity of the findings

No comment

Additional comments

This systematic review is based on efficacy of IL-6 inhibitor Tocilizumab in Covid-19 patients. The authors presented the review based on 16-case control studies gathered from PubMed search. Considering the ongoing pandemic COVID-19 and lack of effective therapies to treat patients affected with severe disease, careful evaluation of clinical data on existing/new anti-viral drugs is important and much needed. However, currently couple of studies are published on efficacy of Tocilizumab in COVID-19 patients as systematic review and meta-analysis (PMID: 32712333, 32713784 & Aziz M et al 2020). The authors can improve the article with following…
1) The introduction part of the review is somewhat lengthy and covered topics which doesn’t have scope in the present review. Authors can present a short and informative introduction that largely talks about IL-6 role in physiology, dysregulation of immune pathways, IL-6 role in COVID-19 pathology, mechanism of Tocilizumab action and known adverse effects (other IL-6 inhibitors). Instead of presenting the topics on other anti-viral drugs, authors can cite relevant existing literature.
2) Authors did not talk about IL-6 levels and other biomarkers (post-treatment levels, if any)?
3) Discussion can be improved by adding points that can help ongoing/future clinical trials to better assess drug efficacy.

---

## Round 0.2 · accepted · Accept

I recommend acceptance of current manuscript.

Reviewer 3 ·

Basic reporting

No comment

Experimental design

No comment

Validity of the findings

No comment

Additional comments

Authors have addressed all the concerns.